# Effects of Music Therapy on Neuroplasticity, Welfare, and Performance of Piglets Exposed to Music Therapy in the Intra- and Extra-Uterine Phases

**DOI:** 10.3390/ani12172211

**Published:** 2022-08-28

**Authors:** Isabella Cristina de Castro Lippi, Fabiana Ribeiro Caldara, Ibiara Correia de Lima Almeida-Paz, Henrique Biasotto Morais, Agnês Markiy Odakura, Elisabete Castelon Konkiewitz, Welber Sanches Ferreira, Thiago Leite Fraga, Maria Fernanda de Castro Burbarelli, Gisele Aparecida Felix, Rodrigo Garófallo Garcia, Luan Sousa dos Santos

**Affiliations:** 1School of Veterinary Medicine and Animal Science, Paulista State University, Street Prof. Dr. Walter Maurício Corrêa w/n, Botucatu 18618-687, São Paulo, Brazil; 2Faculty of Agricultural Science, Federal University of Grande Dourados, Itahum Highway, km 12, Dourados 79804-970, Mato Grosso do Sul, Brazil; 3Grande Dourados University Center, Veterinary Sciences, UNIGRAN, Street Balbina de Mattos, 2121, Jardim Universitário Dourados, Dourados 79824-900, Mato Grosso do Sul, Brazil; 4Animal Science Institute, Department of Animal Nutrition and Pastures, Federal Rural University of Rio de Janeiro, Highway BR 465, Km 07, w/n, Seropédica 23897-000, Rio de Janeiro, Brazil

**Keywords:** BDNF, behavior, brain, environmental enrichment, neurotrophic, productive performance

## Abstract

**Simple Summary:**

Environmental enrichment using music therapy can be used to improve animal welfare. Music, as an enrichment of the environment, is presented as an easy and viable way to remove the sterility of the breeding environment and make it more interesting and attractive. In this study, we aimed to evaluate the effects of auditory environmental enrichment in the pre- and postpartum period of sows on the behavior, performance, and neuro-plasticity of their piglets. Exposure to music in the last 1/3 of pregnancy and farrowing/lactation improved the weight of piglets at birth and weaning. Musical enrichment during pregnancy and lactation was able to cause changes in the piglets’ neuroplasticity and improve their productive performances.

**Abstract:**

The rearing environment of pigs can cause a high level of stress due to the lack of stimuli and the impossibility of carrying out natural behaviors. Music therapy is a way to enrich the environment and promote stress relief. Few studies in swine using environmental enrichers focus on functional benefits, such as stress resilience, improved biological functions, or mental status. The effect of environmental enrichment on neurobiological processes is particularly poorly understood in farm animals. Thus, our study sought to elucidate the influence of music in piglets exposed to music therapy in the intrauterine and extrauterine phase on neuroplasticity, evaluating the levels of brain-derived neurotrophic factor (BDNF). Behavioural responses were also evaluated using fear tests related to stress resilience. The productive performance of these piglets was analysed to relate the possible reduction in stress levels to greater productivity gains. Forty-eight sows were used at 90 days of gestation until the weaning of their piglets. In the gestation phase, the sows were divided into two treatments: control (without music therapy) and music (with music therapy). In the farrowing/lactation phase, the sows were separated into four treatments: control-control (no music in any phase); control-music (music only in farrowing/lactation); music-control (music only during pregnancy); and music-music (music in both reproductive phases). Music therapy did not cause a difference in the BDNF levels of piglets at birth. However, piglets born from sows of the music-music treatment did not show a reduction in BDNF between birth and weaning, unlike the other treatments. Exposure to music in the last 1/3 of pregnancy and farrowing/lactation improved the weight of piglets at birth and at weaning. Musical enrichment during pregnancy and lactation was able to cause changes in the piglets’ neuroplasticity and improve their productive performances.

## 1. Introduction

Environmental enrichment using music therapy can be used to improve animal welfare and has positive behavioral effects and stress relief, as reported in several species (de Jonge et al., 2008; Alworth & Buerkle, 2013; Wiśniewska et al., 2018) [1,2,3]. In addition to its beneficial effects, the use of auditory environmental enrichment brings some solutions to difficulties encountered in classic environmental enrichment with the use of objects, such as dispute for access (Godyn et al., 2021) [4], high expenses with the acquisition of objects, maintenance, and labor. Music, as an enrichment of the environment, is presented as an easy and viable way to remove the sterility of the breeding environment and make it more interesting and attractive.

Music for environmental enrichment needs to be carefully selected, and the genre is a significant factor in this choice (Ciborowska et al., 2021) [5]. Classical music, such as Vivaldi, Bach, and Mozart, relaxing music, and meditation music are most related with health promotion; on the other hand, techno and heavy metal are related to higher stress and heart arrhythmia (Trappe, 2010) [6]. Music has multiple factors: rhythm, frequency, tone, loudness, and sound, which influence how sound waves affect the body. In this sense, in humans and rats, frequencies between 4000 and 16,000 Hz increase dopamine synthesis, which reduces blood pressure via dopamine receptors [6,7,8] (Akiyama and Sutoo, 2011; Trappe, 2010; Nuez et al., 2002). In pigs, Li et al., 2020 [9] observed reduced cortisol secretion as a sign of low stress in environments enriched with music, particularly Mozart played at 60–70 dB.

Despite the growing interest in implementing environmental enrichment programs for farm animals, there are few valid scientific measures to assess their effects [10] (Rault et al., 2018). Most studies on animal welfare focus on the mechanisms associated with changes in behavior, quantifying the time the animal spends interacting with the enriching object or exercising inappropriate behaviors, for example. However, few studies focus on the functional benefits of enrichment through variables such as resilience to stress, improvement in biological functions, or mental status. The effect of environmental enrichment on neurobiological development and neural processes is particularly poorly understood in domestic animals (van de Weerd & Day, 2009) [11]. One way of evaluating neurobiological development is through the evaluation of neurotrophins, such as the brain-derived neurotrophic factor (BDNF; Lu et al., 2014) [12]. 

Neurotrophic factors have the capacity to mediate cellular functions, promoting the activation of receptors and the expression of genes that are linked to the regulation of neuroplasticity and cellular health (Schmidt et al., 2008) [13]. Through a systematic review, Kühlmann et al. (2018) [14] concluded that musical intervention presented positive results related to neuroplasticity, structure and brain neurochemistry, behavior, immunology, and physiology in rodents. Prenatal exposure to music has been identified as being beneficial for fetal development and promoted an increase in cells in the motor and somatosensory cortex in rat pups (Kim et al., 2013) [15]. Higher concentrations of BDNF in brain tissue have been associated with improved cognitive functions (Novkovic et al., 2015) [16] and greater stress resilience (Mosaferi et al., 2015) [17] in studies that used environmental enrichment successfully. The measurement of BDNF in peripheral blood can be performed in serum and plasma using venipuncture. Additionally, BDNF crosses the blood-brain barrier and its levels in serum and plasma have a high correlation with BDNF in the cerebrospinal fluid in rats and swine (Klein et al., 2011) [18] and can provide important information about BDNF changes in the brain in a less invasive way (Pan et al., 1998) [19]. 

Considering the impact of prenatal stress on fetal development, it is believed that improving the mother’s well-being during pregnancy, with the use of music therapy, can lead to positive changes in the offspring. Although mother’s well-being indicators were not measured in this study, this paper refers exclusively to the results on piglets. Therefore, in this study, we aimed to evaluate the effects of auditory environmental enrichment in the pre- and postpartum period of sows on the behavior, performance, and neuroplasticity of their piglets. 

## 2. Material and Methods

### 2.1. Location

The protocol was conducted on a commercial farm, located in the municipality of Dourados, MS, Brazil. The municipality is located at latitude 22°13′18″ S, longitude 54°48′23″ W, and altitude of 437 m. According to the Köppen classification, the region’s climate is Aw, with a rainy summer and dry winter, with an average of 1425 mm of annual rainfall and an average annual temperature of 23.4 °C.

### 2.2. Animals, Treatments, and Experimental Design

This study included 48 pregnant sows of the same genetic background (LWx L, Danbreed-90) with parity between three and six and similar body scores. All sows were at 90 days of gestation and selected from an individual stall gestation system. This work was an observational study between two environmental enrichments. The sows were randomly distributed in the two treatments, being transferred to collective pens (12 sows per pen) in two different rooms with the treatments: control group not exposed to music (C) and group with exposure to regular periods of classical music (M). Stall ventilation and temperature were controlled by curtain management. The pen size was 9 × 3 m with a solid concrete floor. No substrate was provided.

At about 110 days of gestation, the sows were transferred to four maternity rooms, where they were housed in conventional farrowing crates. The crates were 2.25 m long, 90 cm wide for the sows, and 1.60 m wide, with a slatted floor. All crates had an incubator with a heating lamp for the piglets. Stall ventilation and temperature were controlled by curtain management. The females were subdivided into 4 treatments, with 12 sows/treatment: control-control (CC): sows that did not listen to music during the gestation and farrowing/lactation phases; control-music (CM): sows that did not listen to music during the gestation phase but listened during the farrowing/lactation period; music-control (MC): sows that listened to music during the gestation phase but did not listen to it during the farrowing/lactation phase; and music-music (MM): sows that listened to music during the gestation and farrowing/lactation periods.

The sows were fed with commercial rations, formulated to meet the nutritional requirements of each phase based on NRC (2012) [20] data, and fed twice a day during the gestation phase with floor feeding, and fed four times during the lactation phase using a built-in trough for feed delivery in the crates. Sows and piglets had free access to fresh water in automatic nipple drinkers. Piglets had access to feed since birth. 

At birth, the umbilical cord of the piglets was tied, and two days after birth, iron dextran was administrated intramuscularly, teeth were clipped, and the tail was docked. No cross-fostering was carried out. 

All facilities were separated by enough physical space to ensure acoustic isolation between treatments. Measurement to determine the acoustic isolation was carried out in every pen (at each 1 m) and every room (at the entry and the exit). All pens and crates had the same weather conditions, and all management was the same in every treatment (except for the music).

### 2.3. Sound Stimuli

The animals submitted to the treatment with classical music were exposed every day of the week to the musical stimulus, during the entire period of the research, starting the treatment on the first day after the transfer of the sows to the collective pens and ending with the weaning of the piglets (at 21 days of age). The selected songs were Bach symphonies chosen at random, forming a two-hour playlist. There was no repetition of the songs over a period of two hours. The sound intensity remained between 60 and 75 dB, as mentioned by Silva (2016), and this range was measured using the Sound Meter (Abc Apps) Decibelimeter application.

The music was played three times throughout the day, all days of the week, from 09:00 to 19:00 h, for 2 h in a row, with an interval of 2 h between each playback sequence. The total period of music exposition during each day was six hours. No repetition of the songs over a period of two hours and an intermittent period of playing were practices adopted to avoid habituation.

### 2.4. Behavioural Assessments

For the behavioral assessment of piglets, three different tests, always performed in the same order, were carried out. On the day before weaning, at 21 days of age, 10 piglets from each treatment were randomly chosen to perform the tests.

### 2.5. Novel Arena Test

The piglets were individually placed in an isolated test area, three meters long and one meter wide, which did not allow eye contact with other animals. The stay time of each animal in the area was four minutes, whose behavior was recorded using a Nikon D5000 (Serial Number 3181774, Tokyo, Japan) digital camera. The activities observed during the period were: active behaviors (rooting, urinating, defecating, and escape attempts) and inactive behaviors (animal standing or lying down). Animal standing, lying down, urinating, defecating, and escape attempts were considered undesired behaviors since they are fear demonstrations; thus, rooting is a desired behavior that demonstrates the animal is calm and interacting with the new area. 

The behavior was evaluated by the instantaneous sampling method, with observations at 10-s intervals, totaling 24 observations per animal/treatment.

### 2.6. Novel Object Test

After completion of the unknown area test, the piglet remained in the test area, with a purple Swiss ball (55 cm in diameter) being placed loose on the floor. The animal’s behavior was recorded using a Nikon D5000 (Serial Number 3181774) digital camera for four minutes and subsequently analyzed similarly to the previous test, including interaction with the unknown object in the active behaviors. The behavior was evaluated by the focal animal method with observations at 10-s intervals, totaling 24 observations per animal/treatment.

### 2.7. Vocalization

The number of indistinct vocalizations was counted during the testing period of the unknown area and the new object, using audio extracted from the Nikon D5000 (Serial Number 3181774) digital camera. Each vocalization (sound above 70 dB) was computed for all piglets used in the test.

### 2.8. Voluntary Approach Test

After carrying out the tests above, the animal remained in the test area. Then, a collaborator who had not had any previous contact with the animals, wearing the uniform provided by the commercial farm, entered the place, and remained next to the wall, immobile, for a maximum of three minutes. The latency time for the piglet to approach and contact the human was counted. If the piglet did not contact the human, the test was finished. 

### 2.9. Productive Performance

Piglets were individually weighed at birth and 21 days of age (weaning). 

### 2.10. Brain-Derived Neurotrophic Factor (BDNF)

Ten piglets (different ones from the behavior tests) of four sows per treatment were randomly selected, from which 2 mL of umbilical cord blood was collected immediately after birth, using a 3–4-mL disposable syringe and a 24 G hypodermic needle, according to an adapted methodology of Pedersen et al. (2011) [21]. After blood collection, they were identified with numbers on the ears. The samples were transferred to 2-mL tubes with clot activator (BD Vacutainer^®^ Hemogard, Mississauga, ON, Canada) and homogenized. Blood collection of the piglets at 21 days of age was performed in the same 10 piglets per treatment, identified with a collar, with the animal positioned in the supine position, on a veterinary immobilizer trough for puncture of 4 mL of blood from the anterior vena cava using a 5-mL disposable syringe and 21 G hypodermic needle. The samples were transferred to tubes with 4 mL clot activator (Labor Import) and homogenized. On both occasions, the tubes were left at room temperature for one hour to retract the clot. Subsequently, the samples were subjected to the centrifugation process at 1000× *g* for 15 min using the CELM model LS3 Plus centrifuge.

After centrifugation, the serum was collected with the aid of an automatic pipette and placed in an Eppendorf tube, being frozen until analysis in a freezer with a temperature of −80 °C. The measurement of serum BDNF levels was obtained using the biosensis^®^ Mature BDNF RapidTM ELISA kit (Thebarton, SA, Australia): Human, Mouse, Rat, and blood processing was carried out in accordance with the instructions found in the manufacturer’s manual.

### 2.11. Statistical Analysis

Statistical analyses of the behavioral results were performed using the SAS GLIMMIX procedure (SAS, Version 9.4, SAS Institute Inc., Cary, NC, USA). The studied variables were previously tested to meet the assumptions of normality and, when necessary, transformed to a logarithmic scale; the frequencies are always presented (ilink function of the GLIMMIX procedure). Differences between means with *p* < 0.05 were accepted as statistically different for the behavior tests.

The productive performance and the brain-derived neurotrophic factor were analyzed using the R—3.6.2 program.

For analysis of the birth weight and weaning weight, averages of the litter weight of each sow used in the experiment were used. First, descriptive statistics and outlier removal were performed. Then, the results were tested for normality of the residues using the Shapiro–Wilk test (*p* ≥ 0.05) and homogeneity of variances using the Bartlett test (*p* ≥ 0.05). After confirming compliance with the premises, an analysis of variance (ANOVA) was performed and in the case of a significant result, a Tukey test (*p* < 0.05) was performed to compare means. 

For BDNF, the results were tested for normality of the residues using the Lilliefors–Kolmogorov–Smirnov test (*p* ≥ 0.05) and homogeneity of variances through the test of Levene (*p* ≥ 0.05). After confirming compliance with the premises, an analysis of variance (ANOVA) was performed, and in the case of significant results, a Tukey test (*p* < 0.05) was performed to compare means. A repeated measures model was used.

## 3. Results

### 3.1. Behavioral Assessments

#### Novel Arena Test

There were no significant differences (*p* > 0.05) between the behavior of piglets from the CC, CM, MC, and MM treatments during the novel arena test (Table 1).

### 3.2. Novel Object Test

The behaviors rooting, urinating, defecating, standing, and interaction with the object did not differ (*p* > 0.05) among the piglets from treatments CC, CM, MC, and MM (Table 2). Piglets from the CC and MC treatments tried to escape more often than animals from the CM and MM treatments (*p* = 0.0214; Table 2).

### 3.3. Vocalization and Voluntary Approach Test

Piglets from the CM treatment vocalized less than those from the other treatments (*p* = 0.0016; Table 3). There was no significant difference (*p* = 0.6954) in the average time for the voluntary approach of piglets for the CC, CM, MC, and MM treatments (Table 3). 

### 3.4. Productive Performance

The birth weight of piglets descended from sows that were not exposed to music therapy during pregnancy was lower when compared to the group exposed to music (*p* = 0.009). Piglets exposed to music in the prepartum and postpartum phase showed greater weight at weaning when compared to piglets in the CC group (*p* = 0.0150; Table 4).

### 3.5. Brain-Derived Neurotrophic Factor (BDNF)

The concentration of serum BDNF at birth did not differ between treatments (*p* = 0.296). On the other hand, at weaning, the serum BDNF concentration was significantly higher (*p* = 0.0005) in piglets from the MM treatment, in which the animals were exposed to classical music in the stages of pregnancy and lactation (Table 5).

The CC, CM, and MC treatments showed significantly higher BDNF concentrations at birth when compared to weaning. This fact was not verified in the MM treatment, which showed no statistical difference (*p* = 0.5516) between these two phases (Table 5).

## 4. Discussion

### 4.1. Behavioral Assessments

The effects of environmental enrichment (EA) on animals are scientifically proven to be beneficial. However, it remains unclear how it affects fetal development, and little is known about the possible changes in progeny behavior and brain development after mothers’ exposure to EA. The mother’s experiences and the uterine environment can have effects on the offspring. Factors such as emotional reactivity, responsiveness to stressors, and levels of cognition can be modulated by challenges in the prenatal and neonatal periods (Weinstock, 2008; Rutherford et al., 2014) [22,23].

At the end of pregnancy, the fetal brain has functional glucocorticoid receptors, which can be affected by stress experienced by the mother, shaping important brain structures, and generating negative effects [24] (Baxter et al., 2016). Environmental enrichment during late pregnancy can alter the offspring’s phenotype, making them more adjusted to their environment (Tatemoto et al., 2019) [25].

Enrichment for pigs in the growth phase results in a greater ability to adapt to new situations, reduced incidence of negative behaviors, reduced fear, and improved learning ability (Van de Perre et al., 2011; Roy et al., 2019) [26,27].

#### 4.1.1. Novel Arena Test

In general, the behavior of pigs in the situation of an unknown area has a small or no correlation with the behavior in other fear tests and social isolation is the main factor that affects the behavior in this test, especially when group-raised animals are tested individually (Forkman et al., 2007) [28].

Similar to the present study, researchers found no difference between sterile and enriched systems when piglets were exposed to a novel arena test. It was suggested that separation from the sow and exposure to an unknown environment had a strong effect on all piglets, which may have masked possible differences in the stress response between rearing systems (Chaloupková et al., 2007) [29]. 

#### 4.1.2. Novel Object Test

Environmental enrichment leads to a reduction in the fear of piglets and an increase in adaptability to the new situation. Thus, animals spend more time being active in new environments (Jansen et al., 2009) [30]. The use of structural, cognitive, and substrate enrichment stimulates motivation for exploration, greater contact with a new object, and less signs of stress, such as defecation and escape attempts, suggesting an improvement in well-being (Vanheukelom et al., 2012; Zebunke et al., 2013) [31,32]. 

Piglets from the CM and MM treatments showed fewer escape attempts compared to animals that did not listen to music during the maternity period. Possibly, this indicates a decrease in the stress level of these pigs and a beneficial effect of music therapy in reducing neophobic behaviors. Rearing in an environment enriched with music therapy may not have been influential in increasing exploratory behaviors and contact with the new object since the piglets were not used to enrichment objects being introduced in their pens, such as toys, substrate, ropes, or rubbers.

#### 4.1.3. Vocalization and Voluntary Approach Test

Vocalization provides a relevant indication of the level of arousal in response to a new situation, being correlated with plasma adrenaline levels [28] (Forkman et al., 2007). Environmental enrichment reduces the number of piglets’ vocalizations in contrast to offspring reared in sterile environments [29] (Chaloupková et al., 2007). However, in the present study, only the piglets in the CM treatment had less vocalizations while the piglets in the other treatments showed similar results.

A study has shown that piglets from enriched pens showed lesser signs of stress and fear in the voluntary approach test [29] (Chaloupková et al., 2007). However, the nature of a long-term relationship with humans can induce a certain judgment on the part of piglets, indicating that the emotional state of farm animals can be affected by the way humans interact with them and their previous experiences [33] (Brajon et al., 2015). In the present study, although the collaborator in the test was not familiar to the piglets, there was no difference in the approach time of the animals. This fact may be related to the proper management of the team of employees and researchers, who had a positive relationship with the animals. 

#### 4.1.4. Productive Performance

The last 1/3 of pregnancy is linked to the accelerated growth of fetuses, with individual growth rates reaching 4.1 g/d, and development of the mammary gland (McPherson et al., 2004; Trujillo-Ortega et al., 2006) [34,35]. Piglets from the MM and MC treatments had the highest average birth weights. In research by [36] Liu et al. (2016), the authors observed that two weeks of exposure to music reduced the stress and anxiety of pregnant women. Accordingly, Garcia-Gonzalez et al. (2018) [37] observed that music therapy during pregnancy was positively related to the birth weight of babies, concluding that there is a direct relationship between mother’s exposure to music, lower levels of anxiety, and greater newborn weight [37] (Garcia-Gonzalez et al., 2018). High levels of stress and anxiety have been linked to low birth weight of babies [38,39] (Copper et al., 1996; Bhagwanani et al., 1997). 

Similar to the research reported with humans, in the present study, the piglets in the MM and MC treatments had a higher average birth weight, demonstrating the positive effect of music therapy during the gestation phase in relation to the litter weight. We can possibly relate this result to the reduction of stress, confirmed by the lower frequency of stereotypes and agonistic behaviors of sows exposed to musical stimulation in the last 1/3 of pregnancy [40] (Lippi et al., unpublished results).

Rearing in an enriched environment in the pre- and postpartum period reduces stress and increases circulating oxytocin levels, positively affecting piglet growth [41] (Yun et al., 2014), which could justify the best results being obtained in the treatment in which the music was played during pregnancy and maternity when compared to the CC treatment. The worse weight results at weaning of the piglets in the CC group can be justified since pregnancy is naturally a potential stressful phase for sows and, according to Bosch et al. (2007) [42], stress for sows during pregnancy results in decreased time spent in the nest. It is also possible to notice that the sows in the CC treatment had the worst behavioral results during the experimental pregnancy period in a companion paper [40] (Lippi et al., unpublished results). Evaluating pregnant sows, Ringgenberg et al. (2012) [43] concluded that stress during this phase reduces maternal care, reducing responses to piglets’ calls and lactation posture. The mixture of sows in group housing represents a stressful event that occurs during the gestation of sows [44] (Silva, 2016).

#### 4.1.5. Brain-Derived Neurotrophic Factor (BDNF)

Substantial variation in blood BDNF levels was found among piglets in the same treatments as in [10] Rault et al. (2018)’s study, which may reflect a functional variation in the effect of music in each animal, possibly depending on their interaction with sound. Differences in the BDNF concentration may occur due to regular biological changes, which is commonly reported in the literature for other species [45] (Lommatzsch et al., 2005).

The neurotrophic factors mediate cellular functions, promoting the activation of receptors and the expression of genes that are linked to the regulation of neuroplasticity and cellular health [13] (Schmidt et al., 2008). Increased neuroplasticity in mouse puppies from mothers exposed to music therapy was reported by [15] Kim et al. (2013). In chicks, sound stimulation in the prenatal phase increased the expression of the synaptic protein in the auditory nuclei of the brain stem and increased the size and number of neurons in the auditory association area of the anterior brain [46] (Alladi et al., 2002). Likewise, a quantitative analysis of embryos stimulated by sound revealed significantly increased expression of BDNF in the medio-rostral nidopallium and ventral hyperpallium (MNH) areas, which plays a key role in auditory impression [47,48] (Bredenkötter & Braun, 2000; Panicker et al., 2002), although higher BDNF was not verified in the early born piglets in the present study. 

Environmental enrichment during adult life induces changes in behavior and BDNF in the mouse brain, with the hippocampus being one of the most susceptible areas to the effects of the enrichment [49] (Zhu et al., 2006). These results were obtained with brain tissue analysis, unlike the present study, in which the measurement of brain-derived neurotrophic factor was carried out in blood serum. Despite the high correlation of the BDNF levels in serum and plasma with BDNF in the cerebrospinal fluid in rats and swine (Klein et al., 2011) [18], the difference in the methodology used to determine the BDNF concentration and the fact that the sizes of certain brain areas that could have been impacted by music were not measured may justify the disagreement of our results with the identified literature. 

Through repeated sampling in the same individuals, the results show that the BDNF concentration decreased with age in the CC, CM, and MC treatments. This result is expected since the most intense brain development is related to early life and the neurotrophic role of BDNF. Brain plasticity decreases with ageing, as does BDNF in the blood (Lommatzsch et al., 2005) [45]. A decrease in BDNF expression is associated with neuronal atrophy or death that occurs, for example, with ageing or some neurological disorders (Murer et al., 2001) [50].

On the other hand, the MM treatment did not differ between the two sampling phases of the experiment (birth and weaning) and showed a higher mean concentration at weaning when compared to the other groups in this same phase.

Chaudhury and Wadhwa (2009) [51] point out that groups stimulated by sound in the prenatal phase, when exposed to the sound again in the postnatal period, possibly present an improved response, with an increase in CREB mRNA, which regulates the transcriptional activation of the gene BDNF through its phosphorylation. This hypothesis is further confirmed by a study demonstrating that exposure to music in the perinatal period improves learning performance and alters BDNF signaling in adult mice [52] (Chikahisa et al., 2006). These results suggest the facilitation of postnatal synaptic plasticity after prenatal stimulation with music [53] (Roy et al., 2014), which could explain the higher concentration of BDNF in the MM treatment when compared to the others. Although the neurotrophic factor did not increase between birth and weaning in the MM group, music therapy may have inhibited the natural reduction of BDNF expression, leading to the maintenance of its concentration and a higher average compared to piglets in the other treatments at the same age, which showed statistically lower averages at weaning.

Exposure to music during the last 1/3 of pregnancy and lactation improved birth and weaned weight. Exposure to music therapy in the pre- and postnatal periods acted as an inhibitor of the natural reduction of the BDNF concentration with ageing. 

Auditory environmental enrichment is a feasible and simple way to improve the performance and quality of life of the animal, reducing stress and resulting in positive impacts on production. In this sense, reduced levels of BDNF [54] (Phillips, 2017) may be indicative of chronic stress and higher levels of BDNF indicate greater resilience to stress (Mosaferi et al., 2015) [17], as shown in research that used environmental enrichment successfully. Thus, the maintenance of high levels of BDNF in piglets exposed to music therapy may be an indication that this group of animals were less stressed during the period evaluated in this study.

The data from this study demonstrate the benefits of music therapy, evidencing its relaxing characteristics, which may be supported by an improvement in neuroplasticity and zootechnical indices. Measurement of the level of welfare of pigs in production may be a difficult task, but neuroplasticity and animal performance can be used as tools for assessing welfare. Thus, the use of these indicators in the evaluation of animal welfare showed that positive effects of music therapy on the welfare of piglets could be noted.

## 5. Conclusions

Exposure to music in the last 1/3 of pregnancy and farrowing/lactation improved the weight of piglets at birth and weaning. Musical enrichment during pregnancy and lactation was able to cause changes in the piglets’ neuroplasticity and improve their productive performances. Further studies should be carried out to investigate the exposure of sows to music from the beginning of pregnancy and analyze their descendants in the phases after weaning.

## Figures and Tables

**Table 1 animals-12-02211-t001:** Effect of auditory environmental enrichment on the mean proportion of behaviors of piglets distributed in the CC treatments: control-control; CM: control-music; MC: music-control; MM: music-music, during the novel arena test.

Behaviors	Treatment					
	CC	CM	MC	MM	SEM	*p*-Value
Observations (n)	24	24	24	24		
Rooting	0.2900	0.2882	0.3051	0.2625	0.05	0.9216
Urinating	0.0016	0.0065	0.0024	0.0083	0.04	0.7449
Defecating	0.0096	0.0414	0.0154	0.0083	0.01	0.1224
Escape Attempts	0.0750	0.0583	0.0375	0.0750	0.02	0.6228
Standing	0.3417	0.4250	0.3458	0.2708	0.06	0.3008
Lying	<0.0001	<0.0001	0.0417	0.0041	0.02	0.2473

SEM = standard error of the mean.

**Table 2 animals-12-02211-t002:** Effect of auditory environmental enrichment on the mean proportion behavior of piglets distributed in CC treatments: control-control; CM: control-music; MC: music-control; MM: music-music, during the novel object test.

Behaviors	Treatment					
	CC	CM	MC	MM	SEM	*p*-Value
Observations (n)	24	24	24	24		
Rooting	0.1178	0.2226	0.1572	0.125	0.03	0.0887
Urinating	<0.0001	0.0042	0.0139	0.0042	0.01	0.3529
Defecating	0.0125	0.0083	0.0139	0.0125	0.01	0.9692
Escape Attempts	0.1492 a	0.0384 b	0.1508 a	0.0542 b	0.05	0.0214
Standing	0.3212	0.4547	0.333	0.4125	0.05	0.2308
Interaction with object	0.1292	0.0792	0.1833	0.2458	0.05	0.0713

Means followed by different letters in the same row are statistically different according to Tukey’s test at 5% significance. SEM = standard error of the mean.

**Table 3 animals-12-02211-t003:** Latency for a voluntary approach to humans (seconds) and the average number of vocalizations of piglets distributed in the CC treatments: control-control; CM: control-music; MC: music-control; MM: music-music.

Variables	Treatment					
	CC	CM	MC	MM	SEM	*p*-Value
Vocalization (n)	153.90 a	18.09 b	141.78 a	109.70 a	64.19	0.0016
Voluntary approach test (s)	18.10	25.45	10.48	16.64	8.84	0.6954

Means followed by different letters in the same row are statistically different according to Tukey’s test at 5% significance. SEM = standard error of the mean.

**Table 4 animals-12-02211-t004:** Productive performance evaluated in piglets descended from sows distributed in the CC treatments: control-control; CM: control-music; MC: music-control; MM: music-music.

Treatments						
	CC	CM	MC	MM	SEM	*p*-Value
Birth weight (kg)	0.94 b	0.93 b	1.16 a	1.26 a	0.03	0.0009
Weight at weaning (kg)	4.29 b	4.30 ab	4.64 ab	4.75 a	0.06	0.0150

Means followed by different letters in the same row are statistically different according to Tukey’s test at 5% significance. SEM = standard error of the mean.

**Table 5 animals-12-02211-t005:** Concentration of serum BDNF (pg/mL) of piglets at birth and 21 days (weaning), descendants of sows submitted to the CC treatments: control-control; CM: control-music; MC: music-control; MM: music-music.

Treatment	Phase			
	Birth	Weaning	SEM	*p*-Value
MM	59.84 a	56.90 Aa	2.78	0.5516
MC	61.57 a	42.75 Bb	3.18	0.0010
CM	56.67 a	38.61 Bb	4.11	0.0176
CC	55.08 a	38.25 Bb	3.12	0.0050
SEM	3.24	1.51		
*p*-value	0.8270	0.0005		

Means followed by different lowercase letters in the same row or different uppercase letters in the same column are statistically different according to Tukey’s test at 5% significance. SEM = standard error of the mean.

## Data Availability

Not applicable.

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
