# Peer review of "Effects of Music Therapy on Neuroplasticity, Welfare, and Performance of Piglets Exposed to Music Therapy in the Intra- and Extra-Uterine Phases"

_animals, 2022, doi:10.3390/ani12172211_

Round 1
Reviewer 1 Report
Comments to the authors
§ L101: The study included 48 pregnant sows of the same genetic background (LWx L, Danbreed-90) with parity between three and six and similar body scores. All sows were at 90 days of gestation and selected from an individual stall gestation system.
§ L120-121: add appropriate reference (e.g., NRC 2012)
§ L123-125: rephrase the sentence
§ L126-127: iron dextran was administrated intramuscularly??
§ L138: at 21 days of age
§ L144-145: rephrase the sentence
§ L154: add the serial number of Nikon D5000 digital camera.
§ L431-443: check the font size
§ Discussion: you should underline the significance of your results for animal welfare in a separate paragraph.
Author Response
Dear Reviewer,
Thank you for your suggestions.
In order to improve the quality of our manuscript, we seek to respond as best as possible to the suggestions made.
The corrections made are marked in the text in red and indicated in the body of this letter.
Best regards
Reviewer 1
Comments to the authors
- L101: The study included 48 pregnant sows of the same genetic background (LWx L, Danbreed-90) with parity between three and six and similar body scores. All sows were at 90 days of gestation and selected from an individual stall gestation system.
R: Changed in the text.
- L120-121: add appropriate reference (e.g., NRC 2012)
R: reference included
- L123-125: rephrase the sentence
R: Sentence reformulated
- L126-127: iron dextran was administrated intramuscularly??
R: information included
- L138: at 21 days of age
R: Changed in the text.
- L144-145: rephrase the sentence
R: Sentence reformulated
- L154: add the serial number of Nikon D5000 digital camera.
R: information included
- L431-443: check the font size
R: Changed in the text.
- Discussion: you should underline the significance of your results for animal welfare in a separate paragraph.
R: paragraph included

Reviewer 2 Report
Title and in general: I would not call it music therapy because therapy presupposes a disease.
Line 49: How easy and feasible is this method really in practice? The barn must be technically equipped for the music and technology is always susceptible to interference, especially when it is exposed to harmful environmental influences like dust or ammonia in the barn. When cleaning the barn the technic must be protected. I doubt that farmers consider playing classical music to be particularly practical in their barns.
Line 58: Does 4000 to 16000 Hz refer to pigs or humans or does it apply in general?
Line 60: must be Li et al., 2020
Line 89: The hypothesis is that the well-being of the mother during pregnancy has an influence on the piglets. However, the well-being of the sows was not investigated in this study or the results are not published here. It should be clearly written at this point that this paper refers exclusively to the results on the piglets.
Line 107: must be "The pen" with capital T
Line 111: It would be good to know what is the standard for pig husbandry in Brazil, since there are great differences internationally also in terms of legal requirements.
Line 155: A description of how the behaviors were evaluated is missing here. Was active behavior evaluated positively or negatively? What does negative behavior mean (see also line 289)? Please provide a more precise definition and explanation of the evaluation. For example, lying down can certainly indicate relaxation and can be seen just as positively as exploration by the animals.
What does animal rooting mean? The vocabulary should be reconsidered here. It is not clear what behavior is meant by it. Do you mean exploration behavior?
Line 167: Was the type of vocalization distinguished? Pigs can make different sounds and there are also studies on which sounds of the animals are due to positive or negative emotions. Again, there is no explanation of how the vocalizations were evaluated.
Line 178: How many piglets per sow were born? Have there been any mortalities? Were animals sick and had to be treated? I think only the weight as a performance parameter is not enough.
Line 194: Where is Figure 1?
General comment on the experimental design: I think that one run is not enough to evaluate the results in a significant way. At least two litters of the same sows should have been studied, because one litter may be different from the next, the respective lactation of the sow is related to the weight development of the piglets, the number of piglets, the body condition of the sow before birth, etc.
Line 275 ff: What is the Piglet fear test? Where is it explained? I am not clear about the relation of this paragraph to the results.
Line 298: Then this test does not seem to be suitable for achieving significant results. Why was this test selected and performed? What was hoped for when it is already known from other studies that this test is only of limited significance?
Line 322: Must be "less signs"
General comment: The research question is interesting for scientists, but I think that it has little practical relevance for pig breeding. There are many other approaches that can reduce stress and improve animal welfare in practice, which are better tested and are already applied in practice. Usually, such approaches aim to fulfill a natural stimulus or need of the animals to thereby improve the housing environment. The exposure to music is not a physiological need of the animals that they want to fulfill under natural conditions. Therefore I think that other approaches are better. In my opinion, the results are relatively weak and not very meaningful, because only one run was studied per group and a larger number of replicates especially of the same sows is necessary to validate the results. Furthermore, I think that hardly any farmer will be convinced to expose his sows to classical music in the barn. Research on stress reduction, animal welfare and performance improvement should be conducted with the intention of establishing these new methods in practice. However, I do not see this in this study. Nevertheless, it is an interesting theme.
Author Response
Dear Reviewer,
Thank you for your suggestions.
In order to improve the quality of our manuscript, we seek to respond as best as possible to the suggestions made.
The corrections made are marked in the text in blue and indicated in the body of this letter.
Best regards
Reviewer 2
Comments and Suggestions for Authors
Line 49: How easy and feasible is this method really in practice? The barn must be technically equipped for the music and technology is always susceptible to interference, especially when it is exposed to harmful environmental influences like dust or ammonia in the barn. When cleaning the barn the technic must be protected. I doubt that farmers consider playing classical music to be particularly practical in their barns.
R: The comparison was made in relation to other methods of environmental enrichment, in which the producer has to make substantial changes in the barn or frequent acquisitions/replacement of enrichment material
Line 58: Does 4000 to 16000 Hz refer to pigs or humans or does it apply in general?
R: Information added to the text
Line 60: must be Li et al., 2020
R: Changed in the text
Line 89: The hypothesis is that the well-being of the mother during pregnancy has an influence on the piglets. However, the well-being of the sows was not investigated in this study or the results are not published here. It should be clearly written at this point that this paper refers exclusively to the results on the piglets.
R: Information added to the text
Line 107: must be "The pen" with capital T
R: Changed in the text
Line 155: A description of how the behaviors were evaluated is missing here. Was active behavior evaluated positively or negatively? What does negative behavior mean (see also line 289)? Please provide a more precise definition and explanation of the evaluation. For example, lying down can certainly indicate relaxation and can be seen just as positively as exploration by the animals.
What does animal rooting mean? The vocabulary should be reconsidered here. It is not clear what behavior is meant by it. Do you mean exploration behavior?
R: Information added to the text
Line 167: Was the type of vocalization distinguished? Pigs can make different sounds and there are also studies on which sounds of the animals are due to positive or negative emotions. Again, there is no explanation of how the vocalizations were evaluated.
R: Vocalizations were only counted, this information was added to the text
Line 178: How many piglets per sow were born? Have there been any mortalities? Were animals sick and had to be treated? I think only the weight as a performance parameter is not enough.
R: The litter data were considered as the performance of the sow, therefore they were not included in this manuscript. We chose to consider only the weights in the performance of piglets
Line 194: Where is Figure 1?
R: Figure 1 was removed of the text
Line 275 ff: What is the Piglet fear test? Where is it explained? I am not clear about the relation of this paragraph to the results.
R: Removed of the text
Line 298: Then this test does not seem to be suitable for achieving significant results. Why was this test selected and performed? What was hoped for when it is already known from other studies that this test is only of limited significance?
R: We were hoping to see if the music would bring of any benefit to an intensely challenging situation. This was not observed, though.
Line 322: Must be "less signs"
R: Changed in the text

Reviewer 3 Report
Generally an interesting study and some good points raised however some considerations:
Is a simple summary not needed for all submissions to the journal?
line 19 not sure the 'of the species' is needed
line 20 states few studies on this area of research but not sure that is a fair assessment there seems to be a few others on pigs but also quite a few on rodents
lines 46-50 make a lot of claims about limitations of other EE and benefits of music as EE but not citations to back up these claims.
Line 52 needs a review for phrasing/ grammar
Line 55 why is the impact of techno music just a perhaps? is the evidence in conclusive?
line 60 sing should be sign.
Line 104-105 states sows were moved from individual stalls to collective pens, has it been considered the impact of this on stress and behaviour?
Line 143 sound played 6 hours a day 7 days a week throughout, one of the main considerations of enrichment is its novelty, would be good to have some comment on how habituation is avoided here.
Line 153 grammatical issues with whose behaviour was recorded.
Line 150 down- are the three tests always done in the same order? consideration of order effect on the results if so?
Tests lead to at least 11 mins of social isolation- impacts of these are considered in the discussion and could entirely account for the behavioural data seen. Currently cannot see the justification for separating out individual piglets for this testing and the subsequent stress it has clearly caused in some from the results of the study. This needs to be more clearly explained.
Line 179 down, was the BDNF testing on the same 10 piglets as the behavioural data? or separate 10?
suitable conclusions drawn on the weight of the piglets, think due to methodological issues conclusions on behavioural impacts cannot be established.
Author Response
Dear Reviewer,
Thank you for your suggestions.
In order to improve the quality of our manuscript, we seek to respond as best as possible to the suggestions made.
The corrections made are marked in the text in green and indicated in the body of this letter.
Best regards
Reviewer 3
line 19 not sure the 'of the species' is needed
R: Removed of the text
line 20 states few studies on this area of research but not sure that is a fair assessment there seems to be a few others on pigs but also quite a few on rodents
R: Specificized in the text
lines 46-50 make a lot of claims about limitations of other EE and benefits of music as EE but not citations to back up these claims.
R: reference included
Line 52 needs a review for phrasing/ grammar
R: sentence reformuated
Line 55 why is the impact of techno music just a perhaps? is the evidence in conclusive?
R: Grammar mistake, perhaps changed to on the other hand
line 60 sing should be sign.
R: changed in the text
Line 104-105 states sows were moved from individual stalls to collective pens, has it been considered the impact of this on stress and behaviour?
R: this is a normal practice in the commercial farm, it is part of sows management, they were used to this practice.
Line 143 sound played 6 hours a day 7 days a week throughout, one of the main considerations of enrichment is its novelty, would be good to have some comment on how habituation is avoided here.
R: information included in the text
Line 150 down- are the three tests always done in the same order? consideration of order effect on the results if so?
Tests lead to at least 11 mins of social isolation- impacts of these are considered in the discussion and could entirely account for the behavioural data seen. Currently cannot see the justification for separating out individual piglets for this testing and the subsequent stress it has clearly caused in some from the results of the study. This needs to be more clearly explained.
R: information included in the text
Line 179 down, was the BDNF testing on the same 10 piglets as the behavioural data? or separate 10?
R: information included in the text
suitable conclusions drawn on the weight of the piglets, think due to methodological issues conclusions on behavioural impacts cannot be established.
R: We cannot say with absolute certainty that there was an increase in well-being since it is multifactorial, but the results obtained are positive signs, which we tried to demonstrated in our discussion and conclusion.

Round 2
Reviewer 2 Report
All comments have been edited, thank you.
Author Response
Dear Reviewer,
thank you for your coments.
Best regards
Reviewer 3 Report
There are a few typo's/ grammatical errors in the edits that have been made to the paper which would benefit from changing prior to publication
line 328 the sings should be sings I think, however not sure that is the best word choice either maybe less occurrences would be better
line 190- should be different ones from the behavioural tests
line 176- unsure how measure indistinct vocalisations?
Author Response
Dear Reviewer,
Thank you for your suggestions.
All of the R2 suggestions are highlighted in yellow, in the main manuscript.
Best wishes